

# An Automated Technique for Damage Mapping after Earthquakes by Detecting Changes between High-Resolution Images

**Tianyu Ci[1] and Zhen Liu[2, *], Ying Wang[3], Qigen Lin[4], Wu Di[5]**

[1] College of Global Change and Earth System Science, Beijing Normal University, Beijing, China

[2] Center of Information and Network Technology, Beijing Normal University, Beijing, China

[3] Key Laboratory of Environmental Change and Natural Disaster of MOE, Beijing Normal University, Beijing, China

[4] Academy of Disaster Reduction and Emergency Management, Beijing Normal University, Beijing, China

[5] Heilongjiang Geomatics center of China Bureau of Surveying and maping, Harbin, China

* Correspondence: liuzhen@bnu.edu.cn; Tel.: +86-010-5880-7988

**Abstract:** Improving the speed and accuracy of earthquake disaster loss evaluations is very important for disaster response and rescue. This paper presents a new method for urban damage assessments after earthquake disasters by using a change detection technique between bi-temporal (pre- and post-event) high-resolution optical images. A similarity index derived from a pair of images was used to extract the characteristics of collapsed buildings. In this paper, the methods are illustrated using two case studies. Our results confirmed the effectiveness and precision of the proposed technique with optical data of the damage presented using a block scale.

**Keywords:** change detection; earthquake; buildings damage

## 1. Introduction

After an earthquake disaster, obtaining the spatial distribution of earthquake damage information quickly and allotting limited resources to rescue activities is an important approach to reduce loss. Remote sensing (RS) is an efficient tool for obtaining building damage information over short periods (Joyce et al., 2009;Voigt et al., 2007;Fan et al., 2017). RAPIDMAP refers to the use of real-time remote sensing data acquired immediately after an earthquake to reveal the regional influence and extent of the earthquake (Erdik et al., 2011).

The reduction in casualties in urban areas immediately following an earthquake can be improved if the location and severity of damages can be rapidly assessed. Many studies have presented rapid mapping techniques using aerial or satellite images and related geospatial data, and different methods have been developed (Wang and Li, 2015;Reinartz et al., 2013;Klonus et al., 2012;Chen and Hutchinson, 2011;Vu and Ban, 2010). In the literature, several papers have exploited information obtained from the changes between images. Rapid mapping was largely adopted to support the emergency management activities related to the major disasters that have occurred in recent years. For example, the Copernicus Emergency Management Service (Directorate Space) provides a large set of parameters for users to choose to produce rapid mapping.

By using remote sensing images before and after an earthquake, we can effectively assess the post-earthquake damage. Depending on the data, different methods are used. A review presented in Reference (Joyce et al., 2009) described several rapid remote sensing assessment methods, the use of pre-earthquake and post-earthquake remote sensing images, and change detection methods commonly used for identifying damaged buildings.

Due to the passive nature of optical satellite imaging, features or objects extracted from images may vary as a function of sensor type, orbital position, solar illumination, weather condition and the number of pre-processing steps. Therefore, significant challenges exist in developing robust



change detection and classification methods that will approach the accuracy level achieved by
human intelligence.
This article will examine a rapid assessment mapping method based on change detection. This
method can be applied to satellite images and aerial images and has the following advantages:
(1) The method is applicable to a variety of data sources and sensors; and (2) the method can
rapidly achieve an estimated result with close to real-time automation in the aftermath of a
disaster and with random access to data parameters and platforms that are necessary to complete
the assessment. Although using various filtering, or morphological approaches can also achieve
a similar result, this simple and efficient approach agrees with the core spirit of RAPIDMAP, i.e.,
"Rapid."
*1.1. Related Work*
In the context of rapid disaster mapping, visual interpretation-based, change detection-based,
and machine learning-based methods are some of the approaches that have been explored, and
related review articles can be found in Reference (Erdik et al., 2011). Studies of rapid mapping
using synthetic aperture radar (SAR) data are not included in this section.
Boccardo and Tonolo (Boccardo and Giulio Tonolo, 2015) described the use of remote sensing
in emergency mapping for disaster response as well as the limitations of a satellite-based
approach. Different types of remote sensing sensors, platforms, and techniques have been used
to assess the impact and damage caused by earthquakes (Boccardo et al., 2015;Antonietta et al.,
2015;Svatonova, 2015). Schweier and Markus (Schweier and Markus, 2006) explained damage
types of entire buildings and analyzed which geometrical features could be used to interpreted
building damage. Although the observation of building roofs by optical imagery could not
distinguish all types of destroyed buildings (Plank, 2014), a number of results have been
presented after earthquake events and their accuracy were tested with field investigation data
(Kerle, 2010;Booth et al., 2011).
In the literature, several papers have exploited information carried by remote sensing images
for earthquake damage mapping purposes. As a consequence, the most widely used technique
for reliably assessing urban damage is visual inspection and interpretation (Ehrlich et al., 2009).
In the case study of the Bam earthquake, visual damage interpretation (Stramondo et al.,
2006;Saito et al., 2005) based on the European Macroseismic Scale (EMS-98) was carried out
building-by-building, comparing pre-event and post-event images. In Reference (Huyck et al.,
2005), Huyck and Adams used Neighborhood Edge Dissimilarities for citywide damage mapping
with multi-sensor optical satellite imagery. The location and severity of post-earthquake building
damage was determined by spectral changes, edge detection, and texture analysis, as described
in Reference (Adams, 2004). Another approach for damage evaluation based on object (i.e., single
building) recognition was presented in References .
Compared to image gray values, edge, texture and gradient are near-constant features that
are less influenced by time phases. Furthermore, different visual features provide complementary
evidence for image interpretation. For example, the gradient represents the degree of variation of
neighborhood gray values. Structural similarity, first proposed in Reference (Wang et al., 2004),
has already been widely used for evaluating image quality. Based on structural similarity, many
non-gray-value-based and feature-based change detection methods have been proposed. For
example, Reference (Liu et al., 2005) proposed a method based on texture or gradient similarity
validation and Liu (Liu et al., 2012) conducted an image quality assessment using gradient
similarity.
The rest of the paper is organized as follows. Section 2 presents an introduction to the experiment
data. Section 3 describes and evaluates the gradient similarity index. Section 4 presents a
flowchart of the rapid mapping method. In section 5, an experiment utilizing a remote sensing
image from the study area is discussed. Finally, conclusions are drawn in Section 6.

**2. Study Area and Dataset**





In this study, two different types of remote sensing data were used to analyze the convenience,
efficiency: airborne data acquired from Ludian County and satellite data acquired from Yushu
County. We used the same method to calculate the gradient similarity index to assessment
building damage after earthquake with these datasets which have different spatial and spectral
characteristics
2.1. Airborne Data
An Ms 6.5 earthquake shook Ludian County, Yunnan Province (China) on 3 August 2014,
resulting in 3143 injuries, 617 deaths, and 112 missing persons (Xu et al., 2015). This event caused
exceptionally severe damage at the epicenter, near the town of Longtoushan in Ludian County,
and has been selected as our test area. Figure 1 shows the ruined buildings on both sides of the
main road across the town of Longtoushan. As most of the buildings in this area were not
designed to withstand seismic events, they remained vulnerable to ground motion (Xu et al.,
113   2015).

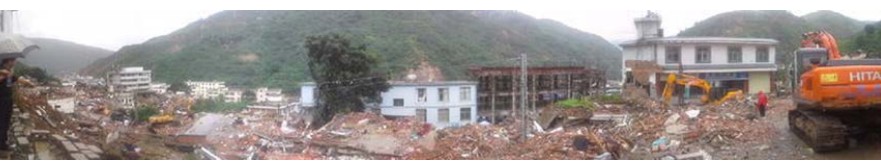


**Figure 1.** Ruined buildings on both sides of the main road across the Longtoushan Town. This photo
sourced from Reference (Xu et al., 2015).
Two aerial images were acquired to map the damage caused by the earthquake in the town of
Longtoushan. The image acquired in 2012 was denoted as the pre-event airborne image, and the
image acquired on 4 August, 2014 was the post-event airborne image for the remainder of the
study. The spatial and spectral characteristics of the two images were the same. These
characteristics both had three spectral bands, R, G and B, and a spatial resolution of 0.2 m. (see
Table 1) The images were geo-referenced and mapped to a cartographic projection. The co-
registered images are shown in Figure 2.

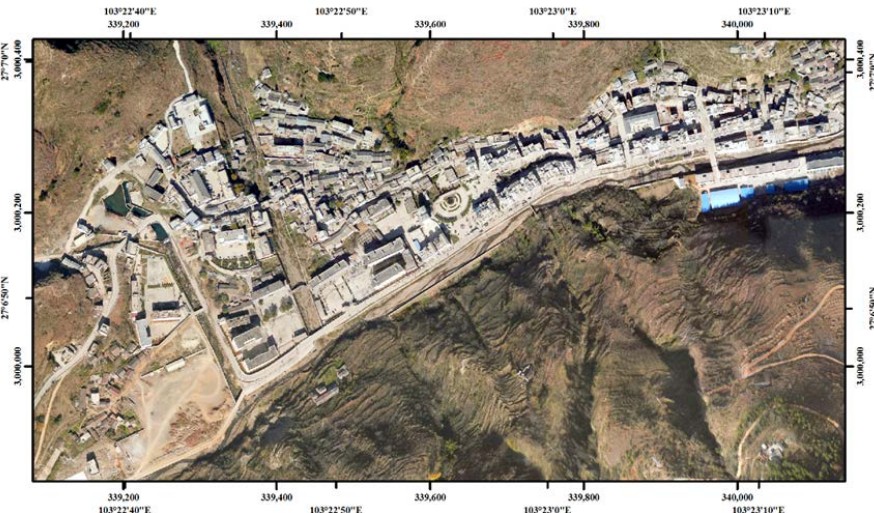






(**a**) The pre-event airborne image of Longtoushan town acquired in 2012.

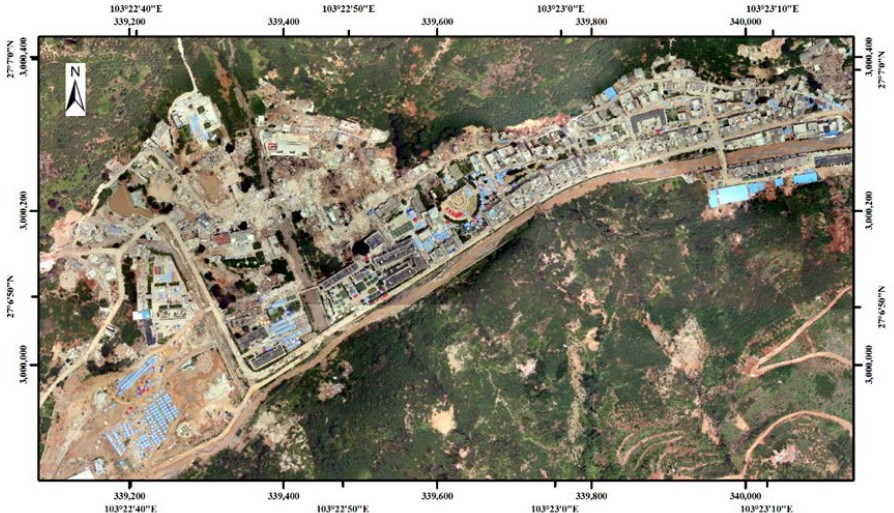


(**b**) The post-event airborne image of Longtoushan town acquired on 4 August 2014, the
day after the earthquake.
**Figure 2.** Pre- (**a**) and post-event (**b**) maps of the experimental area in Longtoushan town
of Ludian in Yunnan.
**Table 1.** The spatial and spectral characteristics of the used data.

| platform | bands | spatial resolution(m) | Acquired time | Location |
|---|---|---|---|---|
| QuickBird | pan | 0.6 | 2010-04-15 | Yushu |
| Ikonos-2 | pan | 0.8 | 2007-11-22 | Yushu |
| airborne | R, G, B | 0.2 | 2014-08-07 | Ludian |
| airborne | R, G, B | 0.2 | 2012 | Ludian |


*2.2. Satellite Data*

137        Yushu County in Qinghai Province, China (geographical coordinates of 31.18N latitude
and 96.78 E longitude) was hit by a 7.1 magnitude earthquake on 14 April 2010. This strong
earthquake caused extensive damage to buildings, facilities, and more than 2000 people were
dead. Fast and reliable information about the location, damage extent and damage level of the
hard-hit areas, particularly urban areas, was important for the rescue planning actions.

142        A post-earthquake QuickBird image from 2010-04-15 (just one day after the earthquake)
and a pre-earthquake IKONOS-2 image which is imaged on 2007-11-22 was used in the study.
For both the pre- and post-earthquake data, a multispectral image with three optical bands and
a near infrared (NIR) band, and a panchromatic image, were available for the analysis. The spatial



resolution of the QuickBird panchromatic image was 0.6 m, and the 0.8 m spatial resolution of
the Ikonos-2 panchromatic image was re-sampled to 0.6 m. (see Table 1). Furthermore, the
radiometric resolution, originally 11-bit, was reduced to 8-bit. All images were projected to UTM
47N and geo-registered. An overview of the images is shown in Figure 3.

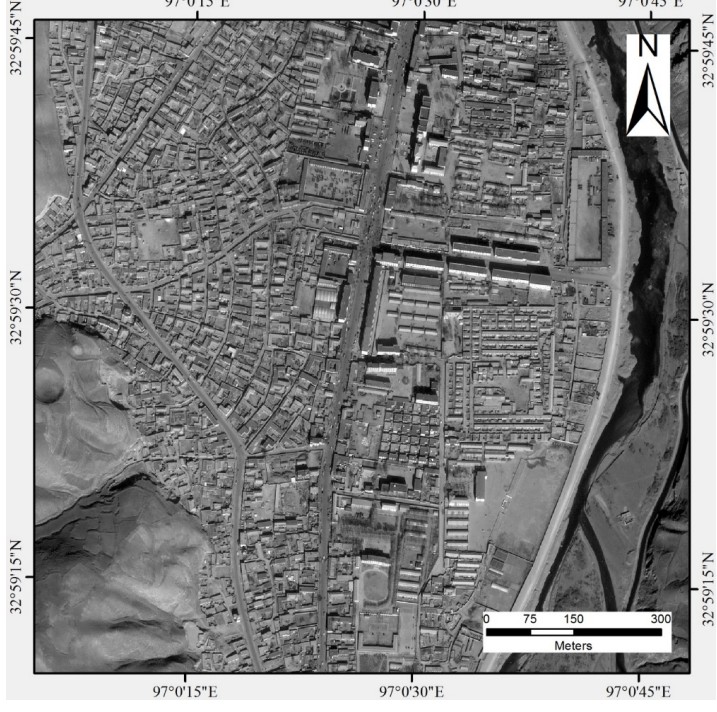

(**a**)




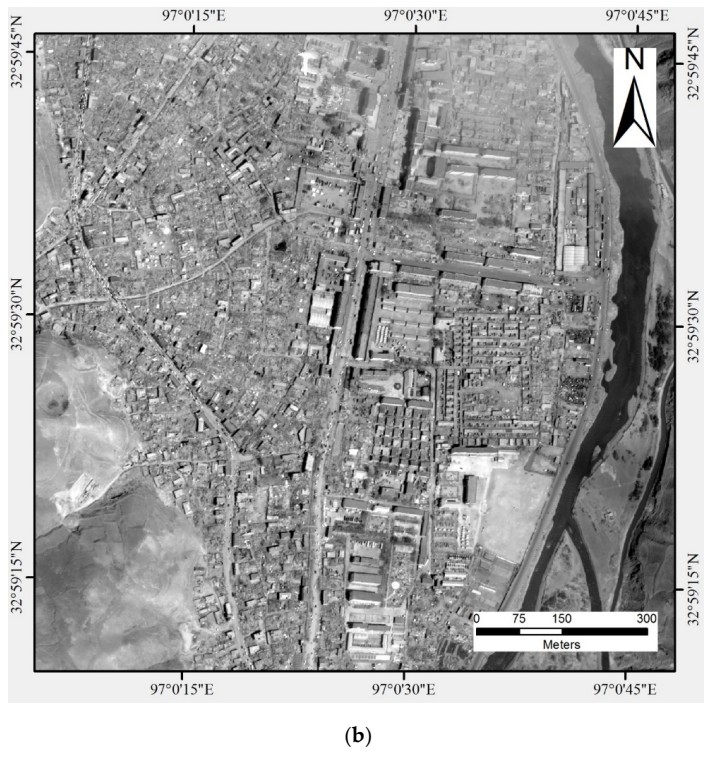

(**b**)

**Figure 3.** Pre- (**a**) and post-event (**b**) maps of the experimental area in Yushu town of Qinghai Province, China.

### 3. Change Detection Using the Gradient Similarity Index

3.1. Structural Similarity Index

The well-cited structural similarity (SSIM) index (Wang et al., 2004), which assumes that natural images are highly structured, has previously been used for evaluating remote sensing image quality and detecting change (Wang, 2010). The structural information in an image is defined as the attributes that represent the structure of the objects in the scene, independent of the average luminance and contrast (Wang et al., 2004).

The structural similarity of two image blocks $x$ and $y$ is defined as follows:

$$l(x,y) = \frac{2\mu_x\mu_y + C_1}{\mu_x^2 + \mu_y^2 + C_1} \tag{1}$$

$$c(x,y) = \frac{2\sigma_x\sigma_y + C_2}{\sigma_x^2 + \sigma_y^2 + C_2} \tag{2}$$

$$s(x,y) = \frac{\sigma_{xy} + C_3}{\sigma_x\sigma_y + C_3} \tag{3}$$

where $\mu_x, \mu_y, \sigma_x, \sigma_y$, and $\sigma_{xy}$ are the mean of image block x, mean of image block y, variance of image block x, variance of image block y, and covariance of block x and block y, respectively. $C_1, C_2$ and $C_3$ are small constants used to prevent the denominator from equaling zero.

The SSIM for the image blocks is given as follows:





$$SSIM(x,y) = [l(x,y)]^\alpha \cdot [c(x,y)]^\beta \cdot [s(x,y)]^\gamma \tag{4}$$

where $\alpha$, $\beta$ and $\gamma$ are positive constants used to adjust the relative importance of the three
components. The higher values of SSIM indicate greater similarity between the image blocks $x$
and $y$.
The schemes in References (Chen et al., 2006;Yang, 2006) were also based on SSIM and
considered the importance of edges. In these schemes, one or more components of the SSIM were
changed to calculate the values in the edge domain (note that the values were calculated in the
pixel domain). For example, the structure comparison component was changed to the gradient
domain, or both the contrast and structure comparison components were modified. In References
(Kim et al., 2010) and (Cheng et al., 2010), a luminance comparison component was not included.
As minor variants of SSIM, these schemes were lacking due to the considerations of the
calibration and registration precision in remote sensing images.
*3.2. Gradient Similarity Index*
With the gradient image computed for bi-temporal images, the gradient similarity index
for a target area can be described as the dissimilarity between structural features. Thus, a
mathematical dissimilarity measure can be obtained to quantify the degree of structural damage.
Mathematically, a dissimilarity measure is a functional that associates a numeric value with a
pair of functions, whose value monotonically varies with a degree of dissimilarity between the
two functions. In our treatment, the gradient similarity index used in this study was defined as
follows:

$$g(x,y) = \frac{2g_x g_y + C_4}{g_x^2 + g_y^2 + C_4} \tag{5}$$

where $g_x$ and $g_y$ are gradient values for the central pixels of image blocks and $C_4$ is the small
constant, shown in Equation (2), that is used to prevent the denominator from equaling zero (e.g.,
0.0001). In addition, $g(x,y)$ is the gradient similarity between x and y and its value lies in [0, 1].
The initial form of the proposed scheme in Equation (5) was mathematically similar to the
luminance/contrast comparison term of SSIM and was more effective than that in the SSIM for
remote sensing image change detection.
The formulation for $g(x,y)$ measures both image contrast (the degree of signal variation)
change and image structure (structure of objects in the scene) change as the gradient value is a
contrast-and-structure variant feature, as demonstrated in Reference (Liu et al., 2012). One may
verify this property by recalling the observed damage in Figure 4; while a homogeneous roof
becomes broken for interior boundaries or cluttered regions, the value calculated by the formula
decreased and vice versa.
Figure 4 illustrates three pairs of buildings extracted from pre- and post-event images; two
of these buildings suffered different levels of structural damage. Among these buildings, the first
building (A) collapsed after the earthquake, where the post-event image of A indicates that the
exterior structural boundary was completely demolished. The complete boundary of the second
building remained intact after the earthquake.
Visual inspection of three example buildings indicated that the structural damage was
primarily characterized by the changes in structural features between the pre- and post-
earthquake images. These structural features included exterior boundaries (structural footprints),
interior discontinuities and homogeneous regions (structural roof) in the pre-event images, as
well as interior boundaries and cluttered regions due to collapse in the post-event images.
Although the afore-mentioned changes in structural features were diverse, they could be




described generally by observing changes in local intensity transitions between the bi-temporal
images.

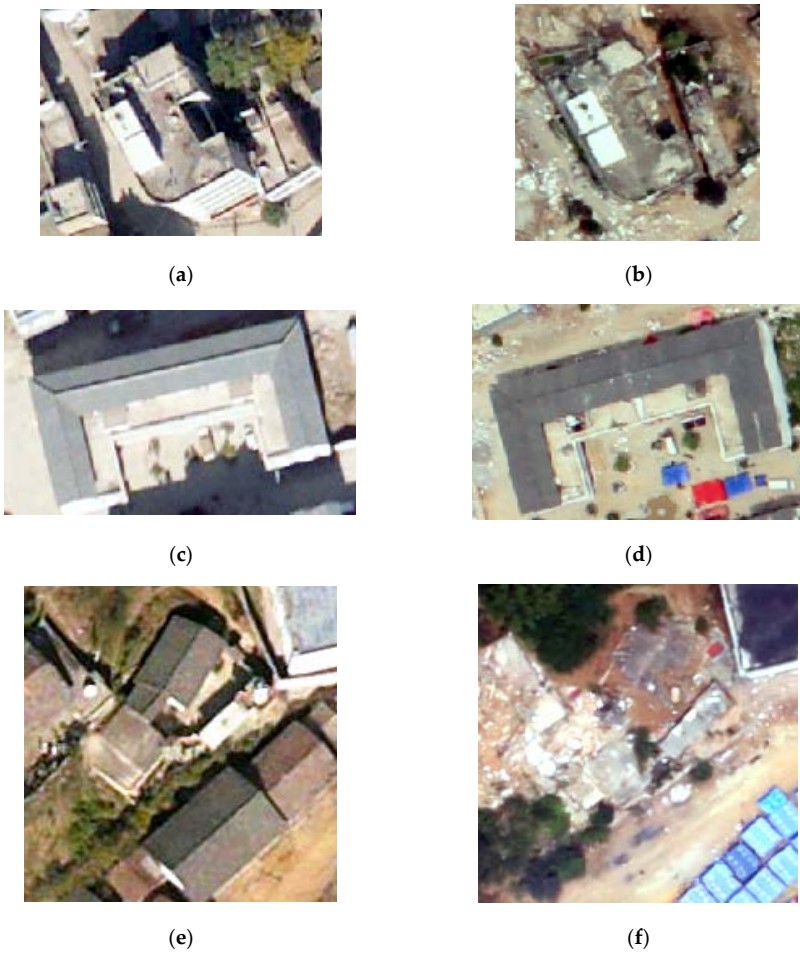

**Figure 4.** Three pairs of example buildings in the pre- and post- images. Images (**a**) and (**b**) are the same building, which was acquired after the earthquake; while (**c**) and (**d**), and (**e**) and (**f**) are the same.

Image gradient magnitudes can be used to amplify grey-level intensity transitions and the
use of image gradients, including magnitude and orientation, is a traditional approach for
extracting image features in computer vision. In addition, many other traditional image features
exist, such as image moments, or co-occurrence texture features. By casting the gradient
computation in the framework of scale-space theory (Bretzner and Lindeberg, 1999), advanced
feature extraction methods such as scale-invariant feature transform (SIFT) have been proposed
to achieve distortion-invariant image features to some degree.
Gradient value was calculated using the Sobel operator (Surhone et al., 2010). In Figure 5,
the resulting image gradient magnitudes corresponding to the pre- and post-event structures in
Figure 3 are illustrated. As observed in both cases, structural features were successfully extracted
with high magnitudes, where significant grey-level intensity transitions occur.



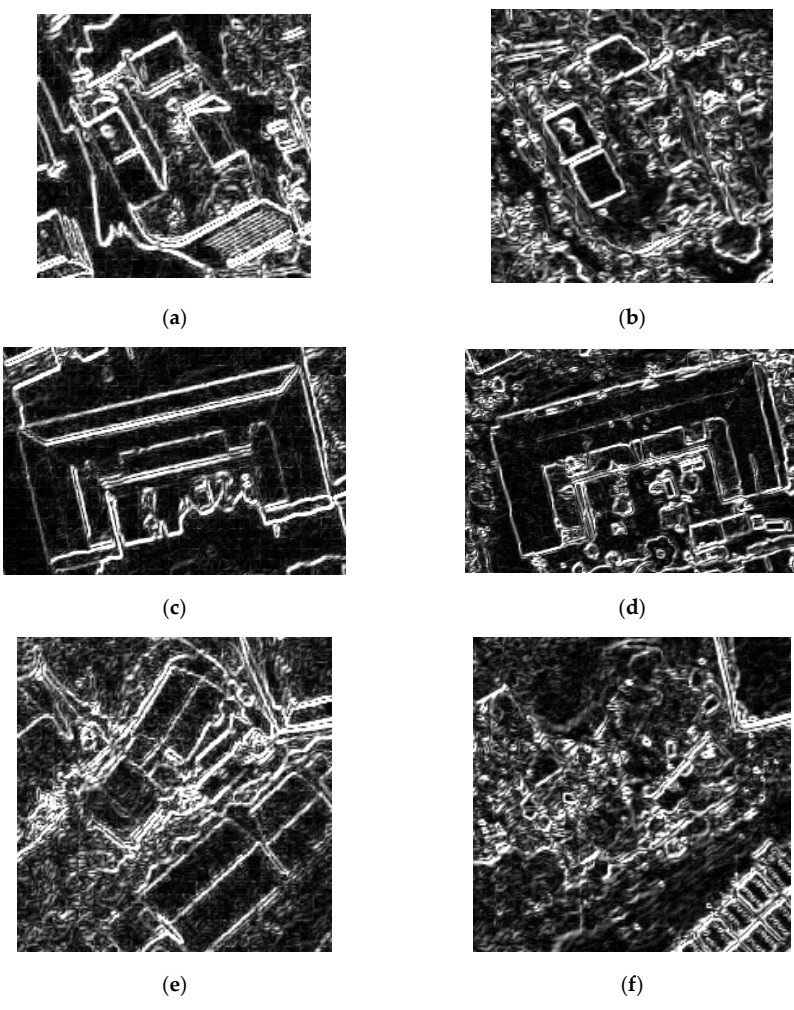

**Figure 5.** Illustrations of gradient images corresponding to the same ID as in Figure 3.

*3.3. Improved Gradient Similarity Index*

Here, we considered that calibration and registration errors occur in remote sensing images. As the formulate g(x, y) is a structure variant feature derived from image gradient features, this method is robust for low precision calibration.

If two images are not in perfect alignment before change detection, the resulting difference image will contain artifacts caused by the incomplete cancellation of unchanged background objects (Ledrew, 1992). These artifacts are referred to as 'registration noise.' One example is given in Figure 6., where the building in Figures 6a and 6b was not in accordance with the footprint of the building. It has been suggested that a geometric correction should result in the two images being within half a pixel of each other (Vu et al., 2005). If this accuracy can be achieved, the registration noise is likely to be less intense than the difference of any real change. However, it is often difficult to keep the geometric correction error below half a pixel for the entire image,



especially in rapid earthquake mapping scenes. The so-called 'standard error', or 'average
residual error' provided by existing geometric-correction software are only estimates from many
individual pixels (ground control points) selected from both images. This type of error inevitably
influences building damage detection in difference images.

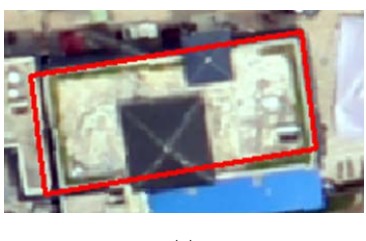 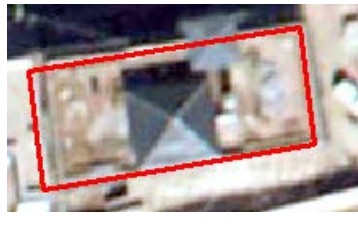

(**a**)      (**b**)

**Figure 6.** An example for registration error in remote sensing images. The building in two image is
the same and should lie in the rectangle if there is no registration error.

In this study, a smoothing-like filter was used for registration-noise reduction. This was
achieved by moving a max-filter over the similarity image and replacing the center pixel with the
maximum value in the moving window. The max-filter function takes a similar form,

$$S'(W) = \max(S(W), S(W'))  \qquad (6)$$

where $S(W)$ is the original gradient similarity, calculated using corresponding pixels from the
pre-event and post-event; $S(W')$ is the candidate gradient similarity, calculated using pixels from
the pre-event image with original position and pixels from post-event with a slide on basis of the
original position; $S'(W)$ is the end result. For example, $S(W)$ was calculated between pixels within
the red rectangle in Figure 7a and pixels within the red rectangle in Figure 7b, while $S(W')$ was
calculated between pixels within the red rectangle in Figure 7a and pixels in a slide rectangle,
such as the green or the blue one. All rectangles whose distance from the original position was
less than the max offset was compared in this formula. The extent over which the gradient
similarity index is calculated can be either a standard mesh grid or an irregular form derived
from image segmentation or a shapefile feature. The max value of the offset is based on the
precision of calibration.

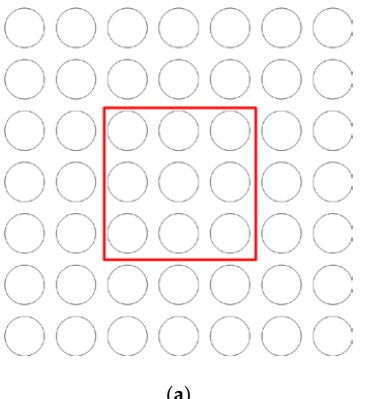 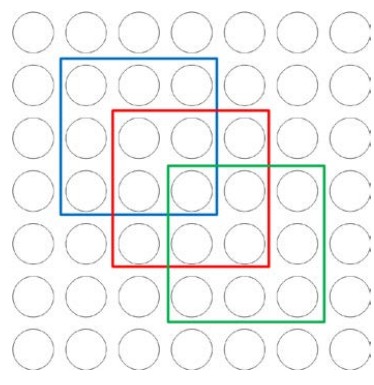

(**a**)      (**b**)





**Figure 7.** Move-window illustrations on a simulating image. (**a**) and (**b**) are both a sub-image of the pre- and
post-event image with the same size and at the same location. Every circle represents a pixel to be processed.
a)
**4. Rapid Mapping Method for Damaged Buildings**
In this study, we present the first results of RAPIDMAP aimed at change detection based
on pre- and post-event optical images. Two pre- and post-event images were used, as shown in
Section 2. The acquisition time of the post-event image was just one day after the earthquake
occurred, therefore we assumed that the destruction caused by the earthquake was captured
completely. Our focus was on the possibility of a rough estimate of damage at a block scale.
A multi-stage earthquake rapid mapping method based on change detection with gradient
similarity was proposed. The general concept of the proposed method can be summarized as
follows. To reduce spectral confusion between buildings and other ground objects (such as intact
buildings and pavements) rather than extracting collapsed buildings directly from bi-temporal
images of the entire study area, buildings and other relevant land-cover types were first extracted
from post-event data using different features and masked. Images of the remaining area were
then used to extract collapsed buildings and conduct rapid damage analysis.
As object-based analysis methods generally outperform pixel-based methods, the
detection of collapsed buildings was implemented at the object level in this study. Specifically,
the proposed method included three successive steps. First, after segmentation of pre-event
images, buildings, pavements (e.g., roads and parking lots), vegetation, and shadows, which
were apparently not collapsed buildings, were extracted using pre-event imagery and masked.
Both spectral and texture images of the remaining area were then classified to extract the
collapsed buildings. Finally, a post-processing step was performed to refine the results obtained.
The procedure of the proposed method is shown in Figure 8.

**Figure 8.** Flowchart for rapid damage detection.
*4.1. Image Segmentation*





As a prerequisite for object-based image analysis, image segmentation was first conducted
to generate appropriate image objects (segments), which were used in the subsequent object-
based extraction of collapsed buildings (Vu et al., 2005;Wang and Li, 2015). Pre-event images
were used in image segmentation to produce consistent objects. The Fractal Net Evolution
Approach (FNEA) algorithm implemented in the eCognition software package—a widely used
multi-resolution segmentation method—was adopted. However, this is not a general
requirement, and any other multilevel segmentation method could have also been used. The
FNEA algorithm is a region-merging technique that starts with each pixel forming one image
object or region. At each step, a pair of image objects is merged into one larger object. The merging
decision is based on local homogeneity criteria, describing the similarity of adjacent image objects.
A 'merging cost,' which represents the degree of fitting, is also assigned to each possible merge.
For a possible merge, the degree of fitting is evaluated and the merge is fulfilled if it is smaller
than a given 'least degree of fitting.' The 'least degree of fitting' value is termed the scale
parameter. The procedure stops when there are no more possible merges.
After image segmentation, the average DN value of pixels within each segment for each
band was calculated to represent spectral features of the segment. The normalized difference
vegetation index (NDVI) average pixel value within each segment was also calculated to
represent the NDVI value of the patch (i.e., object-level NDVI image) and was used to separate
vegetation from non-vegetation.
*4.2. The Difference Image Generation for Damaged Buildings*
The simple assumption made in this study was that if a building was damaged, then its
post-event height would change and the gradient similarity index between pre- and post-events
would be less than the undamaged building. A change analysis was performed to detect the
damaged buildings by the gradient similarity index (GSI) map. The change areas were detected
by a supervised classification method. We can distinguish changed blocks from unchanged
blocks based on a few label data with expert knowledge. We can easily get label data with GIS
software such as ENVI.
Before damage assessment, the vector data were edited as all obtained polygons did not
indicate damaged buildings. Some polygons were vegetation or shadows, thus an NDVI
threshold for vegetation and a mean DN value threshold for shadows was also applied. These
polygons were cleared and polygon regions that may indicate a building region were evaluated.
*4.3. Object-Based Damage Assessment at the Block Level*
Given a pair of bi-temporal satellite images $u0(x, y)$ and $u1(x, y)$, the gradient similarity
index defined in Equation (4) was used to generate similarity measurements at an individual
patch. Due to the underlying monotonic relation between the damage measures and potential
damage levels, simple thresholding or learning-based classification methods can be used to
generate a pixel-wise, binary-level changing stage.
Damage detection means discriminating damaged blocks from undamaged ones. To
accomplish this, the developed approach targets finding debris areas and intact buildings. The
flowchart for scale-space damage detection is illustrated in Figure 8. A leveling transformation
applied to spectral information gradually flattened the image to identify homogeneous regions
across the scale space. The debris areas, in contrast, were represented as the most heterogeneous
areas. Edge information and its texture, therefore, were useful for delineating debris areas. The
identification of possible intact buildings and debris areas were separately processed on the scale-
space prior to the final object-based crosscheck at the scale space. A detailed description of the
processing in Figure 8 is given as follows.



First, the original pre-event image was used to generate an appropriate image block using
image segmentation processing. Vegetation or shadow blocks recognized by NDVI and average
DN values were masked out. Second, gradient similarity index images were calculated using the
Equation provided in Section 3. The final step of damage mapping was to report the damage
situation in an understandable format for stakeholders, such as disaster management
practitioners, earthquake engineers and decision makers. One commonly used damage scale is
the European Macroseismic Scale (EMS), which classifies damage and destruction as heavy
damage, substantial to heavy damage, moderate damage, and light damage. An 'open' approach
was designed here; damage information required presentation as a statistical summary of
damage status for an image object or city block. In addition to maps of building status, it was
necessary to compute the damage area ratio (DAR) for each city block, and to label it with
different damage levels using the flowchart shown in Figure 10.
DAR is computed as:

$$DAR_i = \frac{\sum_i d_{ij}}{A_i} \tag{7}$$

where $DAR_i$ is the DAR value on the ith object polygon; $d_{ij}$ is the "damage flag" (with values 0
or 1) indicating whether pixel j in the polygon was damaged by the earthquake; and $A_i$ is the total
area of the ith polygon.
**5. Results and Discussion**
Following the methodology described in Section 4, extracted damage information from
images is presented and discussed here. Entire data and IDL code used in this research can be
seen on the website: (Ci, 2017).
*5.1 Airborne Data*
In Figure 2, a pair of bi-temporal panchromatic images were shown. The two images, with
a resolution of 20 cm per pixel, were orthorectified and geo-registered. Local spatial alignment
errors between different buildings in the bi-temporal images were frequently found. Hence, the
scale max-filter size was chosen as 2.
Figure 9 shows the damage detection (Figure 9a) and block level assessment (Figure 9b)
results of the rapid damage mapping. Based on image segmentation and block, damage
classification was produced. The National Disaster Reduction Commission of China (NDRCC)
investigated 482 building in Ludian after the earthquake (NDRCC, 2014). In the field
investigation data, 98% of simple structures (brick-wood or civil) were damaged or heavily
affected, and 52.9% of non-simple structure (reinforced concrete) were damaged or heavily
affected. In this experiment, 66% of pixels were identified as damaged.

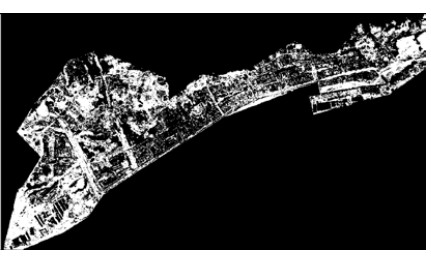

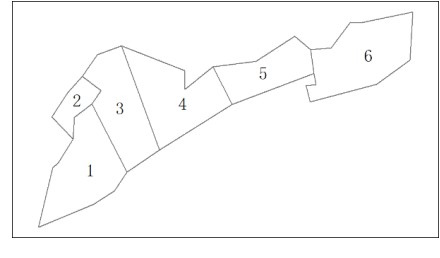

(a)                                    (b)




**Figure 9.** Damage map of Longtoushan town where (**a**) is the change detection result where white represents damaged pixels; and (**b**) shows the id of each polygon, which can be found in Table 3.

This experiment was implemented in the IDL programming language and used about 45 min on a computer with an intel i7 3.4GHz CPU and 12G memory. Efficiency could be highly improved if we rewrote the code with parallel technology.

*5.2. Satellite Data*

We used the same method on the satellite dataset described in Section 2.2. Most buildings to the left of the study area were collapsed (Figure 10). The filter size selected was 2. Buildings on the left side of the image had a high probability of detection as collapsed buildings, which can be easily discovered by visual interpretation. The distribution of the gradient similarity index is drawn in Figure 10d in the two regions with green and blue lines. The gradient similarity index of the green region was obviously bigger than the blue region, and there were more intact buildings in the green region. Damaged buildings in the green region made the distribution of the green line more flat than the blue line. This distribution was also consistent with our opinion in this study.

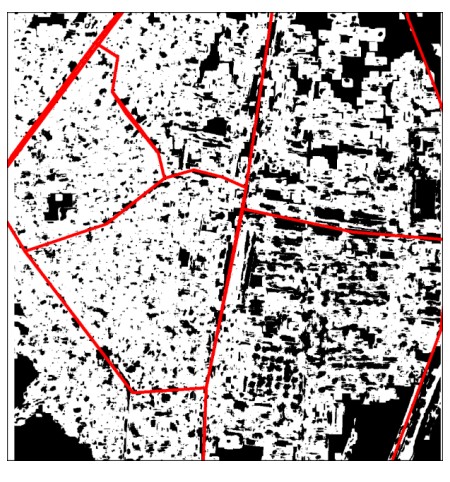
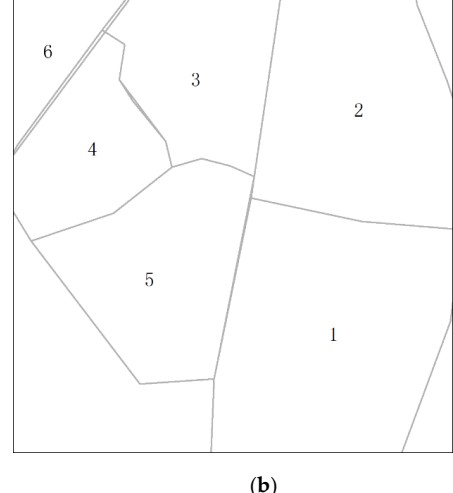

(**a**)  (**b**)



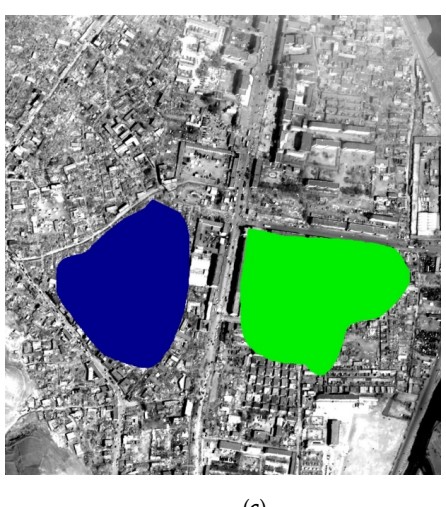

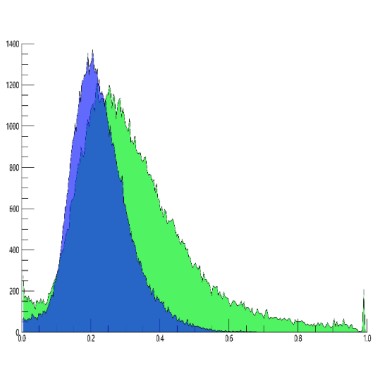

(**c**)                                              (**d**)

**Figure 10.** Damage map of Jiegu town in Yushu, China where (**a**) is the change detection result and white represents the damaged pixels; (**b**) shows the id of each polygon, which can be found in Table 3. (**c**) Mark two regions to be analyzed; and (**d**) in the distribution of gradient similarity index. The green line in (d) represents the GSI distribution of the green region in (c) as well as the blue.

*5.3. Discussion*

We obtained the actual changed buildings by visual interpretation and overlaid it with the change detection results to get the precision statistics of the two experimental areas (see Table 2). To prove the effectiveness of moving distance, we compare the accuracy assessment of different moving distance in the table. From this table, the kappa was very low (around 0.3). The producer's accuracy of intact building was very low. In particular, the low producer's accuracy indicates that many intact areas were wrongly identified as damaged buildings. This may be because the roof of a building cannot represent the footprint of building when the building is high. A more sophisticated method and additional information sources, such as height data pre- and post- event, are required to solve these problems in detail assessment. Different moving distance can influence the accuracy.

**Table 2.** The spatial and spectral characteristics of used data.

| Platform | Moving distance | Overall accuracy | Kappa | Damaged | | Intact | |
|---|---|---|---|---|---|---|---|
| | | | | User accuracy | Producer's accuracy | User accuracy | Producer's accuracy |
| Satellite | 0 | 57.33% | 0.25 | 54.83% | 68.12% | 61.12% | 47.18% |
| | 1 | 61.49% | 0.33 | 58.09% | 68.09% | 64.8% | 55.29% |
| | 2 | 63.63% | 0.37 | 60.96% | 69.51% | 66.94% | 58.09% |
| Airborne | 0 | 53.8% | 0.307 | 84.8% | 64.49% | 76.26% | 45.4% |
| | 1 | 59.17% | 0.26 | 63.31% | 66.25% | 53.16% | 49.94% |



| | | | | | | |
|---|---|---|---|---|---|---|
| 2 | 62.13% | 0.32 | 65.41% | 70.25% | 57.06% | 51.55% |

We also compared the DAR based on the calculated GSI and expert judgement in the study
area. We divided the whole area into six blocks and send the image to four experts who have
worked in damage assessment field for years. They marked the blocks with A-F to represent the
damage intensity from the severe to slight. The average result of experts and the DAR of each
block are shown in table 3. The DAR of each block can be found in figure 9b and 10b. Easy to see,
there is a correlation between DAR and expert judgement.
**Table 3.** The expert judgements for study areas.

| | | City blocks | | | | | |
|---|---|---|---|---|---|---|---|
| | | 1 | 2 | 3 | 4 | 5 | 6 |
| Satellite | Expert judgement | A | C | B | E | D | E |
| | DAR | 58 | 58 | 75 | 84 | 82 | 80 |
| Airborne | Expert judgement | D | E | F | C | A | B |
| | DAR | 85 | 75 | 64 | 56 | 48 | 66 |

Overall, despite the limitations, the comparison shows good agreement for a quick
estimation of damage intensity distribution, especially considering the focus was to produce
geospatial products as a matter of urgency, based on the earliest available images. It also
demonstrated the importance of these products as effective complements to on-going relief
efforts. While the automated damage indication map cannot replace "in depth" damage
assessments, and nor is that the intention; the aim is rather to provide a preliminary (but reliable)
indication of damage distribution for initial disaster relief operations.
**6. Conclusions**
This approach overcomes some issues that often occur in rapid damage assessment
scenarios: first, perfect matching of the images is not required as small shifts can be
accommodated through object linking; second, data from different VHSR sensors can be
compared; and third, parameterization of the rule set and final processing can be performed
sufficiently fast to be used in an operational context.
Finally, it should be emphasized that the automated approach presented herein was not
designed to extract absolute values concerning damaged buildings, nor is it able to completely
replace manual interpretation. Its strength lies in the ability to extract information rapidly (if the
methodological assumptions hold true), thereby assisting users and manual interpreters to
quickly obtain an impression of the spatial distribution of damage in emergency situations and
provide a guide for further, more detailed, analyses.
**Acknowledgments:** This study is supported by the China National Science and Technology Major Project
entitled "The application demonstration system of emergency monitoring and evaluation of major natural
disasters" (03-Y30B06-9001-13/15).
**Author Contributions:** Tianyu Ci and Zhen Liu conceived and designed the experiments; Tianyu Ci, Qingen
Lin did the imagery processing and data analysis; Tianyu Ci, Zhen Liu, Qi Wen and Ying Wang analyzed
the results; Ying Wang and Zhen Liu revised the paper; and Tianyu Ci wrote the paper.
**Conflict of Interest:** The authors declare no conflict of interest.



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
