# Peer review of "Tianyu Ci1 and Zhen Liu2, \*, Ying Wang3, Qigen Lin4, Wu Di5"

_Natural Hazards and Earth System Sciences, 2018_

## Referee Comment (RC1) · Anonymous Referee #1 · 8 May 2018

This paper aims to develop a new method to quickly assess building damage after major disasters. The topic of this paper met the scope of NHESS well and the structure is clear. However, I have a few concerns as listed below: 1) The abstract is too simple, it does not give what is novel (new) in theory and/or methods. Please contain at least one sentence for objective/hypothesis, purpose/aim, experimental material, method of investigation, data collection, result and conclusions. 2) Please modify the flow-chart to include key algorithms of this proposed method. Also, check the typo:" Image processing". 3) The results and discussion should be parallel and divided into two separate sections. 4) The language may be improved by a native speaker, as some sentences sounds awkward. Eg. page 4 line 139-140, page 12 line 305, page 13 line

353, et al.

Specific comments: Page 13 line 351, where is the link? Cannot find either here nor in the references. Page 14 lines 368-369. This sentence is strange. I would wonder why not write your program that way? Figure 10. Why not combine (a) and (b), as (b) has very few information to show. Also consider (d) as an inset of (c).

———————————————————

---

## Referee Comment (RC2) · Anonymous Referee #2 · 27 May 2018

This paper works on the earthquake damage assessment from the pre and post event optical imagery, with the purpose of proposing an automatic and efficient method for quick damage mapping. However, this article is very weak in terms of the innovativeness of the method, the logical rigor of the argument, the clarity of the description, and the usefulness of the discussions. The details are as follows;

(1) The title of this article is "Automated Technique for Damage Mapping after Earthquakes by Detecting Changes between High-Resolution Images", this title does not provide as original information of the article, because damage mapping from the pre and post event optical image using the change detection method is very basic and old

method in this field. One of the keywords of the title is automated technique, unfortunately, I didn't find any demonstration in your paper regarding the degree of automation, did you implemented your method in software or you provide a Pipeline platform for running your method?

(2) Regarding the abstract, the keywords of the abstract are improving the speed and accuracy, unfortunately, this was also not reflected in the manuscript. In terms of speed, how can you say its high speed? What exactly the time you need to perform the post-event damage assessment? How can you say your method is faster than the others? In terms of the accuracy, an overall accuracy of about 60% for block scale damage assessment is too low, especially for the high-resolution optical image? How can the author say it is high accuracy? Therefore the abstract is not consistent with the contents of your manuscript.

(3) The introduction part only give the very basic summary of the previous traditional research, I didn't see the trends and progress of the state of art research in using the optical high-resolution image for damage mapping, the description of the introduction does not provide valuable information for me. In the final part of the introduction, the author mentioned that "The method is applicable to a variety of data sources and sensors", this description is too general, the author also mentioned that rapidly near real time, unfortunately, it is not reflected in the manuscript.

(4) In the related work, the author does not summarize the progress and trends in this field, for example, what are exactly the problems and challenging? What are the trends? How does the previous work inspire you to conduct this research? What's new in your research? The author needs to rewrite this part and the introduction part.

(5) In the study area and dataset part, what is the reason you use high-resolution airborne data, because for me the damage mapping from the high-resolution airborne image is easy, it is quite easy for me to see the difference in Figure 2 visually. What is the purpose of using two cases for comparison? What is the ground truth data for the

damage buildings? How many damage levels of the ground truth data? Where did you obtain the ground truth data? The acquired time in line 120 is not consistent with the Table 1, please correct it. In addition, it will better to provide the geographic information in the map of the study areas? Why did you choose this area for research? Are there any other affected areas that covered by the optical image?

(6) The organization of part3 is unclear for me, In my personal understanding, Structural Similarity Index and Gradient Similarity Index are the previous methods, based on the previous method, your proposed Improved Gradient Similarity Index and applied this method to perform the damage mapping. However, it is quite difficult for me to know what is the inspiration for the author to propose this modification, how this modification can bring effect for damage mapping? What is the theoretical basis for this modification? Why is it important?

(7) Part 4 focus on Rapid Mapping, unfortunately, I did not see the mapping result, the processing step in the figure8 is quite basic, the author did not provide the details of the processing parameters, for example, what size is the block scale? Why you chose this scale? The author mentioned that "As object-based analysis methods generally outperform pixel-based methods", I think the author should give the reference? The author also mentioned that "buildings, pavements (e.g., roads and parking lots), vegetation, and shadows, which were apparently not collapsed buildings, were extracted using pre-event imagery and masked", my question is if the difference between intact buildings is apparently with the collapsed buildings, then what is the significance of your research? .

(8) Part 4.2 mentioned that "The simple assumption made in this study was that if a building was damaged, then its post-event height would change and the gradient similarity index between pre- and post-events would be less than the undamaged building", this assumption does not make sense for me. (9) The DAR is proposed by the other researcher, please cite the publication, as there is no information about your ground truth data, it is difficult for me to know what is the damaged block. (10) In part

5, where is the mapping? How can we know it is rapid or not? Why did you say it is high accuracy?

Please also note the supplement to this comment:
https://www.nat-hazards-earth-syst-sci-discuss.net/nhess-2018-73/nhess-2018-73-RC2-supplement.pdf

---

## Referee Comment (RC3) · Anonymous Referee #3 · 29 May 2018

The paper needs major revisions for publishing in the journal. A native speaker must check the language. Detecting changes is a well established method and I would like to see more the innovative part in this article. New ideas must be mentioned and the introduction is not sufficient enough. It is only a text for previous work. What about the accuracy of this method? Did you use ground control points? I have also some remarks. Line 25 remote sensing is an efficient tool, please correct, it is not a tool. Line 37 reference is missing, line 101 the verb is to assess not to assessment, line 217 reference is missing, line 218 please add the reference of Irwin Sobel 2014. line 227 what does it mean low precision calibration? Section 3.3 how do you improve the gradient similarity index? Figure 8 image processing and not porcessing. This article

must be rewritten to clarify the innovative part.

---

## Author Comment (AC1) · 2 Jul 2018

Dear reviewer:

Thank you for your comments concerning our manuscript entitled "An technique for rapid damage mapping after earthquakes by detecting changes between high-resolution images". Those comments are all valuable and very helpful for revising and improving our paper, as well as the important guiding significance to our researches. We have studied comments carefully and have made correction which we hope meet with approval. Revised portion are marked in red in the paper. The responds to your comments are following:

[Figure]

1) The abstract is too simple, it does not give what is novel (new) in theory and/or methods. Please contain at least one sentence for objective/hypothesis, purpose/aim, experimental material, method of investigation, data collection, result and conclusions.

Responds :We have revised the abstract.

2) Please modify the flowchart to include key algorithms of this proposed method. Also, check the typo:" Image processing".

Responds :We have modified the flowchart

3) The results and discussion should be parallel and divided into two separate sections.

Responds :Done

4) The language may be improved by a native speaker, as some sentences sounds awkward. Eg. page 4 line 139-140, page 12 line 305, page 13 line 353, et al.

Responds :We have modified.

Specific comments: Page 13 line 351, where is the link? Cannot find either here nor in the references.

Responds :We have modified. We add the link and change the references.

Page 14 lines 368-369. This sentence is strange. I would wonder why not write your program that way?

Responds :We have modified.

Figure 10. Why not combine (a) and (b), as (b) has very few information to show. Also consider (d) as an inset of (c).

Responds :We have modified.

Special thanks to you for your good comments.
* * *

---

## Author Comment (AC2) · 2 Jul 2018

Dear reviewer:

Thank you for your comments concerning our manuscript entitled "An technique for rapid damage mapping after earthquakes by detecting changes between high-resolution images". Those comments are all valuable and very helpful for revising and improving our paper, as well as the important guiding significance to our researches. We have studied comments carefully and have made correction which we hope meet with approval. Revised portion are marked in red in the paper. The responds to your comments are following:

A native speaker must check the language.

ResponseïijŽwe have used a service of English editing.

Detecting changes is a well established method and I would like to see more the innovative part in this article. New ideas must be mentioned and the introduction is not sufficient enough. It is only a text for previous work. What about the accuracy of this method? Did you use ground control points? I have also some remarks.

Response :The accuracy was discussed in section 6. We did not use ground truth as we don't have that data.

Line 25 remote sensing is an efficient tool, please correct, it is not a tool.

Response: Done

Line 37 reference is missing, line 101 the verb is to assess not to assessment, line 217 reference is missing, line 218 please add the reference of Irwin Sobel 2014.

Response: Thank you for this comments. We have modified them.

Line 227 what does it mean low precision calibration? Section 3.3 how do you improve the gradient similarity index?

Response: We have delete the old line 227. In section 3.3, A smoothing-like filter was used for registration-noise reduction.

Figure 8 image processing and not porcessing.

Response: we have modified the figure.

Special thanks to you for your good comments.

---

## Author Comment (AC3) · 4 Jul 2018

Dear reviewer:

Thank you for your comments concerning our manuscript entitled "An technique for rapid damage mapping after earthquakes by detecting changes between high-resolution images". Those comments are all valuable and very helpful for revising and improving our paper, as well as the important guiding significance to our researches. We have studied comments carefully and have made correction which we hope meet with approval. Revised portion are marked in red in the paper. The responds to your comments are the additional Word file:

(1) The title of this article is "Automated Technique for Damage Mapping after Earthquakes by Detecting Changes between High-Resolution Images", this title does not provide as original information of the article, because damage mapping from the pre and post event optical image using the change detection method is very basic and old method in this field. One of the keywords of the title is automated technique, unfortunately, I didn't find any demonstration in your paper regarding the degree of automation, did you implemented your method in software or you provide a Pipeline platform for running your method?

Response: We published the code on github and the reference can be found page 13 line 352. We modify the title of this article. Although it is a basic method, we want to try some different. In section 3.3, we use a smoothing-like filter and Gradient Similarity Index for registration-noise reduction

(2) Regarding the abstract, the keywords of the abstract are improving the speed and accuracy, unfortunately, this was also not reflected in the manuscript. In terms of speed, how can you say its high speed? What exactly the time you need to perform the post-event damage assessment? How can you say your method is faster than the others? In terms of the accuracy, an overall accuracy of about 60% for block scale damage assessment is too low, especially for the high-resolution optical image? How can the author say it is high accuracy? Therefore the abstract is not consistent with the contents of your manuscript.

Response: In section 5.2, we show the time used in the experiment. In section 6, the we compare our result and expert judgements. We also have comments in section 7.

(3) The introduction part only give the very basic summary of the previous traditional research, I didn't see the trends and progress of the state of art research in using the optical high-resolution image for damage mapping, the description of the introduction does not provide valuable information for me. In the final part of the introduction, the author mentioned that "The method is applicable to a variety of data sources and sensors", this description is too general, the author also mentioned that rapidly near real time, unfortunately, it is not reflected in the manuscript.

Response : We used two different dataset with the method to prove that. The speed can be found in section 5.2.

(4) In the related work, the author does not summarize the progress and trends in this field, for example, what are exactly the problems and challenging? What are the trends? How does the previous work inspire you to conduct this research? What's new in your research? The author needs to rewrite this part and the introduction part.

Response: we have modified these parts.

(5) In the study area and dataset part, what is the reason you use high-resolution airborne data, because for me the damage mapping from the high-resolution airborne image is easy, it is quite easy for me to see the difference in Figure 2 visually. What is the purpose of using two cases for comparison? What is the ground truth data for the damage buildings? How many damage levels of the ground truth data? Where did you obtain the ground truth data? The acquired time in line 120 is not consistent with the Table 1, please correct it. In addition, it will better to provide the geographic information in the map of the study areas? Why did you choose this area for research? Are there any other affected areas that covered by the optical image?

Response: We used two different dataset to prove the method could be applied on both of them. We used visual interpretation instead of ground truth. We have modified the Table 1. We provide coordinate on the map in figure 2.

(6) The organization of part3 is unclear for me, In my personal understanding, Structural Similarity Index and Gradient Similarity Index are the previous methods, based on the previous method, your proposed Improved Gradient Similarity Index and applied this method to perform the damage mapping. However, it is quite difficult for me to know what is the inspiration for the author to propose this modification, how this modification can bring effect for damage mapping? What is the theoretical basis for this modification? Why is it important?

Response: The two index are both previous methods. We used them in the field of damage assessment. We also do modification to get rid of registration-noise.

(7) Part 4 focus on Rapid Mapping, unfortunately, I did not see the mapping result, the processing step in the figure8 is quite basic, the author did not provide the details of the processing parameters, for example, what size is the block scale? Why you chose this scale? The author mentioned that "As object-based analysis methods generally outperform pixel-based methods", I think the author should give the reference? The author also mentioned that "buildings, pavements (e.g., roads and parking lots), vegetation, and shadows, which were apparently not collapsed buildings, were extracted using pre-event imagery and masked", my question is if the difference between intact buildings is apparently with the collapsed buildings, then what is the significance of your research?

Response: we add some reference for object-based methods. The difference is apparently for human beings, but we want to find method for computer.

(8) Part 4.2 mentioned that "The simple assumption made in this study was that if a building was damaged, then its post-event height would change and the gradient similarity index between pre- and post-events would be less than the undamaged building", this assumption does not make sense for me.

Response: We have some example for gradient similarity index difference on damage buildings in section 3.2.

(9) The DAR is proposed by the other researcher, please cite the publication, as there is no information about your ground truth data, it is difficult for me to know what is the damaged block.

Response: we add reference for DAR. The damage block is shown in figure 10(b) and

[Figure]

figure 9(b).

(10) In part 5, where is the mapping? How can we know it is rapid or not? Why did you say it is high accuracy?

Response: we get the DAR of city blocks in table3.

Special thanks to you for your good comments.
* * *